# Organic multicomponent microparticle libraries

Dandan Zhang[1], Jianbo De[1], Yilong Lei [1✉] & Hongbing Fu [1,2✉]

Multimetallic nanostructures can be synthesized by integrating up to seven or eight metallic elements into a single nanoparticle, which represent a great advance in developing complex multicomponent nanoparticle libraries. Contrary, organic micro- and nanoparticles beyond three π-conjugated components have not been explored because of the diversity and structural complexity of molecular assemblies. Here, we report a library of microparticles composed of an arbitrary combination of four luminescent organic semiconductors. We demonstrate that the composition and emission color of each domain as well as its spatial distribution can be rationally modulated. Unary, binary, ternary, and quaternary microparticles are thus realized in a predictable manner based on the miscibility of the components, resulting in mixed-composition phases or alloyed or phase separated heterostructures. This work reports a simple yet available synthetic methodology for rational modulation of organic multicomponent microparticles with complex architectures, which can be used to direct the design of functional microparticles.

[1] Department of Chemistry, School of Science, Tianjin University, Tianjin, P. R. China. [2] Beijing Key Laboratory for Optical Materials and Photonic Devices, Department of Chemistry, Capital Normal University, Beijing, P. R. China. ✉email: yilonglei@tju.edu.cn; hongbing.fu@tju.edu.cn

Multicomponent nanostructures are formed by integrating multiple metals into a single nanoparticle are emerging as an important class of functional nanoscale architectures, that show a variety of applications ranging from catalysis[1–8], plasmonics[9–12], to biological imaging[13–18]. In these complex nanoparticle architectures, diverse metal components are joined together in an alloyed or phase separated state, thus leading to multiple heterogeneous interfaces at the atomic or nanometer scale. The generated interfaces among different components facilitate unique synergistic coupling effects, giving chemical and physical properties that are superior to pure components. For this reason, great efforts have been made and several breakthrough results have been achieved by developing libraries of multimetallic nanoparticles exceeding three metal components[19–21]. Quinary nanostructures involving five different metals (Au, Ag, Cu, Co, and Ni) were constructed via scanning probe block copolymer lithography (SPBCL), in which alloyed and/or heterostructured phases coexist in one particle depending on the miscibility of each combination of the five components.[19]. Coincidentally, regardless of the miscibility of the component combination, high-entropy-alloy nanoparticles with up to eight elements were also obtained by an unconventional two-step carbothermal shock (CTS) method[21].

Compared to inorganic analogs, organic microparticles and nanoparticles made of multiple π-conjugated molecules have also drawn considerable attention due to their promising performance in photonics and electronics applications[22–39]. Until now, relevant studies in this context mainly focus on binary mixtures that exist in an ordered (host–guest complexes[22–24], cocrystal[25–29]) or random stacking fashion (alloy phase[30–33]) or in a phase separated state[34–39]. For instance, assemblies made of alloys of two structurally similar oligo-phenylenevinylenes (OPVs) exhibit tunable lasing action upon adjusting the stoichiometric ratio between them[40]. Furthermore, two-block or three-block tubular heterojunctions were reported by successive crystallization of fluorinated analogs on non-fluorinated graphite-like seed nanotubes, and application as nanoscale electronic devices, such as diodes and transistors were demonstrated[41]. Similar to the cases of crystalline molecular monolayers[42–46], the mixing behavior of these binary assemblies depends on the miscibility of the components, which is governed by the shape, size and symmetries of the molecules. In contrast, the synthesis of organic multicomponent particles exceeding three molecules remains an unsolved problem due to the greater compositional and structural complexity. Despite the difficulties, however, the abilities to synthesize binary molecular assemblies make it possible to achieve complex multicomponent particles by combining together three or more π-conjugated molecules with adjustable miscibility in a systematic and progressive fashion.

Herein, we construct organic microparticle libraries by combining up to four organic semiconductors into one particle via solution-based assembly routes, using either one-pot or seeded-growth. Depending on the miscibility of the organic molecules, organic multicomponent assemblies with different mixing behavior formed in a rational and predictable manner (Fig. 1). For binary microparticles, alloyed or heterostructured configurations were achieved, which can be used to direct the synthesis of ternary and quaternary microparticles with higher structural complexity. Importantly, the spatial distribution, composition and emission color of each domain for ternary and quaternary microparticles can thus be precisely predicted and modulated based on the binary data. The rational construction of organic multicomponent microparticle libraries provides an ideal platform to study alloy formation and phase separation, which may pave the way in developing multifunctional photonic and electronic devices.

## Results

**Design principle for microparticle libraries of four organic semiconductors**. We selected four prototype π-conjugated organic semiconductors (Fig. 1a), *p*-distyrylbenzene (**1**), tetracene (**2**), 9,10-bis(phenylethynyl)anthracene (**3**), and 5,12-bis(phenylethynyl)naphthacene (**4**) as building blocks to study the assembly of the compound combinations in view of the following considerations: (i) These crystalline materials display adjustable visible emissions ranging from blue to red, which makes them promising candidates for a variety of optoelectronic applications such as lasers[47], optical waveguide[48–50], and light-emitting transistors[51]. (ii) The constituent materials with different structural compatibility enable the construction of multicomponent assemblies with diverse phase behavior. For instance, we found that the combination of **3** and **4** can assemble into binary alloy ribbons with variable composition[30]. Moreover, the material combination of **1** and **2** which show a similarity in molecular configuration and size can also form two-component alloys by substitution of **1** with a small molar ratio of **2**[52,53]. While the combination of compounds **1** and **3** or **1** and **4**, may result in heterostructured or isolated particles in their phase separated state due to their obvious structural differences.

To test this hypothesis, we studied the cases of binary **3/4** and **1/3** assemblies formed via one-pot co-assembly route by tracking the real-time evolution processes (Supplementary Movies 1 and 2). For **3/4** assemblies, a stock solution of **3/4** in THF ($x_4 = N_4/(N_3 + N_4) = 10\%$) was mixed with an ethanol/$H_2O$ (v/v = 4:1) mixture to induce their co-assembly. Under UV excitation, one can observe that a short rod with uniform red emission would emerge at the initial stage ($t \leq 5$ s) and then elongates with during time. (Fig. 2a). In case of the **1/3** combination, a ribbon composed of **1** crystallizes preferentially within 1 min after, which a stepwise growth of multiple short rod-shaped branches forms upon slow evaporation of a solution of **1** and **3** in THF/cyclohexane (Fig. 2d). Observing the growth of the binary assemblies in real time (Supplementary Movies 1–4), we found that the four organic components adopt the following growth sequences: $\mathbf{1} \approx \mathbf{2} \gg \mathbf{3} \approx \mathbf{4}$ (Fig. 2b, e), thus revealing the miscibility of diverse binary material combinations. In particular, compound combinations **1/2** and **3/4** are completely miscible while the combinations **1/3** and **2/3** show partial miscibility. In contrast, compound combinations **1/4** and **2/4** have incomplete miscibility.

The mixing behavior of these binary assemblies can be well understood, however, the prediction of material combinations with up to three or four components is more challenging due to the increased structural complexity. Despite such difficulties, we still expect to achieve all combinations of the ternary and quaternary particles based on the information of the binary systems. To access such multicomponent architectures, solution-based assembly routes involving one-pot and/or two-step seeded growth method were proposed. The present design principle allows us to synthesize and study each combination of multicomponent assemblies made of **1**, **2**, **3**, and **4** in a rational and controlled manner (Fig. 1b–e).

**Unary microparticles from four organic semiconductors**. To examine the feasibility of the design principle for multicomponent assemblies, we first synthesized the simplest assemblies, i.e., pure organic microparticles via solution-based crystallization strategy. Typically, assemblies of **1** were obtained by slow evaporation of a solution of **1** in THF/cyclohexane. Similarly, assemblies of **2** were obtained by fast mixing a monomer solution of **2** in THF with cyclohexane, while assemblies of **3** or **4** were formed by mixing a monomer solution containing **3** or **4** in THF with ethanol/$H_2O$ mixture, respectively.

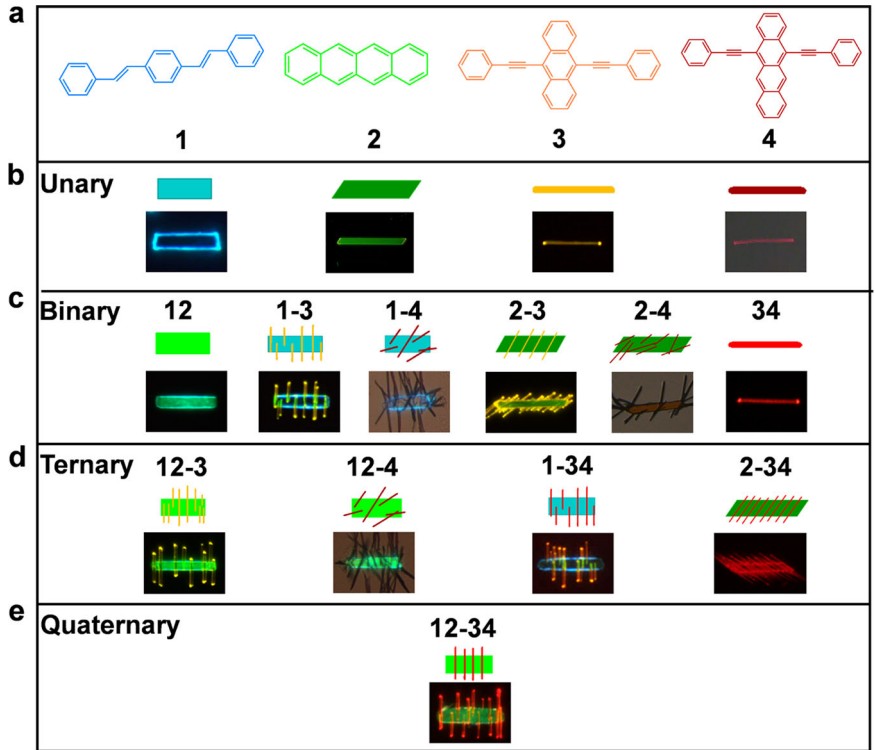

**Fig. 1 Microparticle libraries of four organic semiconductors from unary to quaternary microparticles formed via solution-based assembly routes.**
**a** Schematic showing the molecular structures of four organic semiconductors involving **1** (blue), **2** (green), **3** (orange), and **4** (red). **b** Unary microparticles (top row is a color-coded diagram of the expected particle; bottom row is a fluorescence microscopy image of the resultant particle). The particles **1**, **2**, and **3** were excited by UV light, whereas the particle **4** was excited by green light. **c** Binary microparticles made of two-component combination of the four semiconductors. Among them, both **12** ($y_2 = 1\%$) and **34** ($x_4 = 0.5\%$) represent binary alloys; **1–3**, **1–4**, **2–3**, and **2–4** represent binary heterostructures. **1–4** was excited by UV light and imaged in bright-field mode, whereas the remaining microparticles were excited by only UV light. **2–4** was imaged in bright-field mode. **d** Ternary microparticles made of three-component combination of the four semiconductors. **e** A quaternary microparticle made of **1**, **2**, **3**, and **4**.

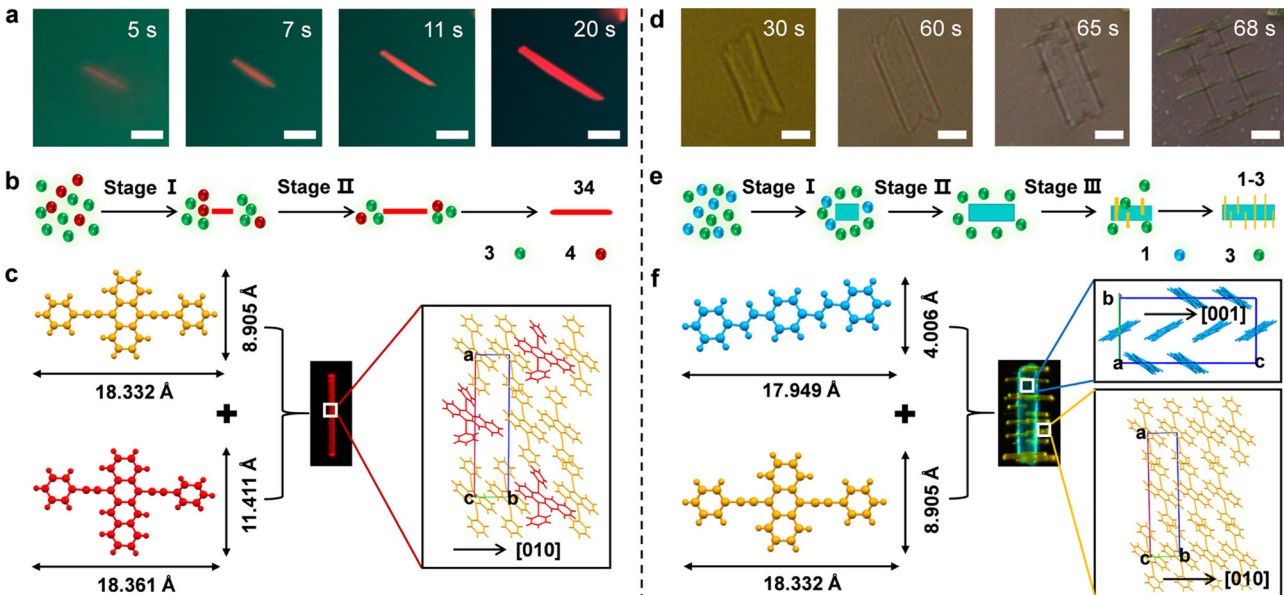

**Fig. 2 The mixing behavior of binary material combinations involving 3/4 and 1/3. a**, **d** The detailed growth processes of single binary microparticles involving **a** 34 ($x_4 = 5\%$) ribbon and **d** 1–3 branched heterostructure at different intervals (**a**, $t = 5$, 7, 11, and 20 s; **d**, $t = 30$, 60, 65, and 68 s). Scale bars, 10 μm. **b**, **e** Schematic showing the mixing behavior of (**b**) 34 ribbons and (**e**) 1–3 branched heterostructures. **c**, **f** The molecular packing motifs of single **c** 34 ribbon along [010] direction and **f** each domain in 1–3 branched heterostructure along [010] direction.

SEM and TEM images (see Supplementary Figs. 1 and 2) of the four unary assemblies reveal that **1** and **2** assembled into ribbon-like appearances, while **3** and **4** were grown as rod-like morphologies. Notably, each of the four micrometer sized particles has a regular appearance and smooth surface. The corresponding powder X-ray diffraction patterns (PXRD, Supplementary Fig. 3) display only few strong diffraction peaks and match well with those simulated from the single-crystal X-ray diffraction data (SCXRD, Supplementary Table 1), indicating their high crystallinity, which is further corroborated by the selected area electron diffraction patterns (SEAD, see Supplementary Fig. 2).

For comparison, the molecular packing motifs and intermolecular interactions of **1**, **2**, **3**, and **4** single crystals (Supplementary Fig. 4) were also revealed based on their crystallographic data. Interestingly, all four crystals adopt a typical herringbone structure with herringbone angles of 59.9°, 51.4°, 79.2° and 84.8°, respectively. The molecules in the crystal structure of **1**–**3** are planar, while the molecules in the crystal structure of **4** appear non-planar with two similar torsion angles (29.5° and 32.5°) between the phenylethynyl group and the naphthacene core. Moreover, there are strong multiple C–H···π interactions in all four crystal structures as well as additional π–π stacking interactions in the crystals **3** and **4**. Combined with the growth morphologies (Supplementary Fig. 5) simulated by the crystallographic data, we infer that individual microparticles of **1**, **2**, **3**, and **4** were grown along [010], [100], [010], and [100] directions, respectively.

Upon UV excitation, the fluorescence microscopy results reveal that the unary assemblies from **1**, **2**, and **3** emit blue, green, and orange light, respectively, while single **4** tube shows dark red light when excited by green light (Fig. 1c). To examine the detailed optical properties, absorption, and photoluminescence (PL) spectra of the four monomer solutions and their assemblies were recorded. Supplementary Fig. 6a depicts multiple resolved absorption peaks for each solution. When excited by UV light, the PL spectra of the above four solutions correspond to blue, cyan, green, and orange light, respectively (inset in Supplementary Fig. 6b). By contrast, the PL spectra of all the crystalline assemblies show obvious red-shifts due to ordered molecular packing and strong intermolecular interactions in each microparticles (Supplementary Fig. 6c). The detailed photophysical parameters of the four components were also summarized in Supplementary Table 2.

**Binary microparticles from four organic semiconductors**. As a proof of concept, we achieved six material combinations of binary microparticles based on the miscibility of diverse two materials. For the completely miscible binary systems, we selected **34** as a typical example and studied its morphology transformation over a wide composition range ($x_4 \leq 50\%$). Specifically, a series of **3/4** stock solutions with variable molar ratio were injected into ethanol/$H_2O$ (v/v) mixtures to induce their co-assembly. As described previously[30], the binary microparticles would undergo structural transformation from straight rods to twisted ribbons upon increasing the content of **4** (Supplementary Fig. 7). As a result, their emission colors were tailored from yellow-green to dark red light (Supplementary Fig. 8a) when excited by blue light, as further verified by the PL spectra (Supplementary Fig. 8b). Moreover, their PXRD patterns would slightly shift with the appearance of new diffraction peaks (Supplementary Fig. 9). The above results clearly reveal the alloy nature of **3/4** co-assemblies. Instead of using physical vapor deposition as previously reported[52,53], we also modified the synthetic procedure of **12** alloys at $y_2 = 0.1\%$, 1%, 10% ($y_2 = N_2/(N_1 + N_2)$) by a drop-casting method. Similar to the case of **34** alloys, binary **12** alloys were also achieved (Supplementary Fig. 10a), as revealed by the corresponding fluorescence microscopy, PL, and PXRD results (Fig. 1c and Supplementary Fig. 11). Thus, **1**- or **3**-based alloys can be constructed based on complete compatibility between **1** and **2** or between **3** and **4**.

Besides the homogeneous alloy assemblies, heterogeneous topological configurations were also realized upon integrating two molecules with partial compatibility. As described above, branched heterostructures made of **1** and **3** have been achieved via one-pot co-assembly route (Fig. 2d). To facilitate the control of branched nanorods, a stepwise seeded growth method was proposed where the pre-existing **1** microribbons serve as seeds and **3** as a second growth unit. Similarly, branched heterostructures comprising a trunk and multiple branches were also formed regardless of one-pot or stepwise seeded growth method (Fig. 2d and Supplementary Fig. 10b). The corresponding PL spectrum shows an almost identical peak profile to that of pure **1** ribbons (Supplementary Fig. 12), probably because **3** has a much smaller PLQY than **1** (Supplementary Table 2). Under UV excitation, the underlying **1** ribbon shows strong blue light, whereas horizontally aligned **3** nanorods attached to the ribbon emit yellow light (Fig. 1c). Only diffraction peaks of **1** are visible in the PXRD pattern (Supplementary Fig. 13), indicating that lattice planes of the **1** ribbon structure are more exposed than those of the **3** rod structure. Additionally, we recorded microarea PL spectra of each domain of **1**–**3** to examine the detailed optical information (Supplementary Fig. 14). When excited by a 405 nm laser, the PL bands of location **i** matches well with blue emission of **1**, while those of location **ii** at a rod tip corresponds to yellow emission of **3**. And the PL spectra of location **iii** at the junction can be regarded as a sum of those of locations **i** and **ii**, further revealing the local composition of **1**–**3** heterostructures.

Naturally, we also examined the case of binary **2/3** system with partial miscibility using **2** microribbons as seeds and **3** as the second growth component. As expected, **3** nanorod arrays were grown on a **2** microribbon, giving branched-like configuration (Supplementary Fig. 10e). Upon excitation with UV light, the **2** ribbon emits green light while the **3** rods emit yellow light (Fig. 1c). We also revealed that a material combination comprising **1** and **4** with incomplete compatibility leads to disordered heterostructured configuration where **4** rods were randomly deposited onto a **1** ribbon (Fig. 1c and Supplementary Fig. 10c). Similarly, we found that disordered nanorod heterostructures were also achieved through stepwise growth of **4** rods on a **2** ribbon (Fig. 1c and Supplementary Fig. 10f). In short, binary heterostructures with aligned or random nanorods were realized based on partial or incomplete structural compatibility of binary material combinations.

**Growth mechanism of binary microparticles with different architectures**. To understand why such different configurations form in these binary systems, we took **34** ($x_4 = 5\%$) and **1**–**3** as typical examples and studied the detailed structural relationship between **3** and **4** or between **1** and **3**. As described above, it has been clearly revealed that compounds **3** and **4** were both arranged in a slipped π–π packing pattern (Supplementary Fig. 4c, d), due to their highly similar π-conjugated molecular skeletons. Moreover, the molecular structure of **3** and **4** has a nearly identical length ($L_3 = 18.332$ Å; $L_4 = 18.361$ Å) and relatively small differences in width ($W_3 = 8.905$ Å; $W_4 = 11.411$ Å), thus facilitating the substitution of **3** by **4** into **34** alloys[30,31]. By means of the crystallographic data of **3**, we simulated the molecular packing mode of **34** alloys and found that **4** molecules would randomly occupy the lattice sites of **3** host (Fig. 2c). However, steric

hindrance of the twisted phenylethynyl groups of **4** would force the parallel stacking of between the anthracene core of **3** and the naphthacene core of **4** to distort at a tilted angle, which enables the maximization of their π–π interactions. At higher molar ratio of **3/4** ($x_4 \geq 33.3\%$), large torsion stress drives the shape transition of **34** alloy assemblies from straight rods to helical ribbons. Moreover, the PL spectrum of **3** shows a large overlap with the absorption spectrum of **4**, as presented in Supplementary Fig. 15a. Well-matched energy levels as well as suitable intermolecular distances between **3** and **4** enable high-efficiency energy transfer (ET) process in **34** alloy assemblies, leading to tunable emissions toward **4**[30]. Notably, compounds **1** and **2** also possess similar molecular packing patterns and effective energy level matching (Supplementary Fig. 15b), giving rise to color-tunable **12** alloy assemblies[52,53].

For **1**–**3** branched heterostructured system (Fig. 2f), **1** and **3** with planar molecular configurations have comparable lengths ($L_1 = 17.949$ Å; $L_3 = 18.332$ Å) and distinctly different widths ($W_1 = 4.006$ Å; $W_3 = 8.905$ Å). As a result, **1** and **3** molecules in their respective assemblies are stacked along [010] direction through C–H···π and π–π interactions (Supplementary Fig. 4a, c and Fig. 2f), respectively, providing a great possibility to achieve such phase separated structures rather than homogeneous alloys. Notably, **3** nanorods were horizontally arranged on **1** ribbon into highly ordered arrays, indicating their small lattice mismatch. The lattice mismatch between **1** ($d_{1, 001} = 15.717$ Å) and **3** ($d_{3, 010} = 5.357$ Å) is calculated to be 2.25%, thus driving epitaxial growth of **3** on the **1** seed ribbon. While the lattice mismatch of the binary combination comprising **1** ($d_{1, 001} = 15.717$ Å) and **4** ($d_{4, 100} = 10.113$ Å) is calculated to be 3.62%, larger than that of **1** and **3**. For the material combination of **1** and **4**, poor structural compatibility forces the molecules of **4** in an arrangement of weakly ordered rods. In the same way, the binary combination comprising **2** ($d_{2, 010} = 7.641$ Å) and **3** ($d_{3, 204} = 3.713$ Å) forms branched nanorod heterostructure ($f = 2.8\%$), whereas that comprising **2** ($d_{2, 010} = 7.641$ Å) and **4** ($d_{4, 021} = 14.812$ Å) produces ill-defined architectures ($f = 3.06\%$). Thus, it can be clearly revealed that different architectures of binary combinations are closely related to their structural compatibility, which are determined by the molecular shapes, sizes, packing motifs, and intermolecular interactions.

**Ternary microparticles from four organic semiconductors**. The ability to synthesize binary microparticles comprising **1**, **2**, **3**, and **4** inspires us to develop multicomponent microparticles toward higher compositional diversity and structural complexity (e.g., ternary microparticles). However, it represents a great challenge to achieve this task due to the difficulties in tuning the growth dynamics of the components, suppressing dissolution-recrystallization of the seeds, and controlling the deposition sequence of each domain. Compared to binary microparticles, the compatibility of the compound combinations in ternary microparticles would become more sophisticated, which can be divided into three types. (I) Two molecules are miscible with each other but the third is partially miscible with one of them and weakly miscible with another. (II) Two molecules are miscible with each other but the third is partially miscible with either. (III) Two molecules are miscible with each other but the third is weakly miscible with either.

On the basis of the binary data, we can infer the possible architectures of complex ternary microparticles (Supplementary Fig. 16a–d). As a case study, we proposed a modified seeded growth strategy to access type I microparticles from **1**, **3**, and **4** (Fig. 3a), in which **1** still acts as a seed and the binary mixture of **3/4** as the second growth unit. Typically, the pre-existing **1**

microribbons were immersed in a precursor solution of **3/4** with variable $x_4$ and then the suspensions were slowly dried. To ensure ordered alignment of **34** rods, $x_4$ should not be higher than 50% due to incomplete compatibility of **1** and **4**. Not surprisingly, a branched-like architecture consisting of a ribbon trunk and multiple branches (Supplementary Fig. 17a) was still achieved as a result of lattice matching between **1** and **34** (Fig. 2f). Under this process, **34** nanorod branches were formed in an initial stoichiometric ratio. As a result, the composition of **34** nanorods as well as emission colors can be tailored by tuning $x_4$ in the precursor solutions. Under UV excitation, **3** ribbon still shows blue light but **34** alloy rods present tunable emissions ranging from yellow to dark red upon increasing $x_4$ (Fig. 3b and Supplementary Fig. 17b). Notably, disordered helical nanobelts rather than aligned nanorods are attached on **1** ribbon at $x_4 = 50\%$ because of large lattice mismatch between **1** and $3_{0.5}4_{0.5}$.

Moreover, microarea PL spectra of single ternary heterostructures formed at $x_4 = 0.01\%$ (Fig. 3c) and 5% (Fig. 3d) were also examined to study luminescent behavior of each domain. When excited by a 405 nm laser, locations **A** and **A′** exhibit similar PL bands in the blue region, while location **C′** shows obvious red-shifted PL peaks relative to those of location **C**, further revealing the composition dependence of PL of each domain. In particular, the blue-emitting domain corresponds to **1** ribbon, whereas the orange-emitting and red-emitting domains correspond to **34**. Moreover, the ternary **1**–**34** heterostructures show similar PXRD pattern to pure **1** or **1**–**3**, because of more exposed crystal facets of **1** ribbon than those of **34** rod (Supplementary Fig. 17c). Hence, it seems reasonable that the ternary type I microparticles architectures are affected by structural compatibility and composition of the compounds.

Coincidentally, we also investigated the case of type II combination of **1**, **2**, and **3** utilizing the preformed **12** alloy ribbons as seeds and **3** as the second growth component. As expected, aligned **3** nanorods were horizontally arranged on the **12** ribbon to form a branched-like architecture (Fig. 1d). The composition and emission colors of the underlying **12** ribbon can be varied by embedding different content of **2** into **1** host ($y_2 = 0.1$, 1, and 10%), whereas yellow emission of **3** rods almost keeps invariable (Supplementary Fig. 18a). The corresponding PL spectra and PXRD patterns further confirm that **12** ribbons serve as seeds to afford ordered growth of **3** rods (Supplementary Fig. 18b, c). Similarly, ternary combination of **2**, **3**, and **4** can also undergo stepwise growth to form type II branched heterostructures (Fig. 1d), in which green-emitting ribbon corresponds to **2** and red-emitting rods correspond to **34**. Moreover, it can be predicted that type III heterostructures with ill-defined morphology (Fig. 1d) will be formed upon employing the **12** ribbons as seeds, and **4** as the second component due to incomplete compatibility of **1** and **4**.

**Quaternary microparticles from four organic semiconductors**. Naturally, we further exploited the library of multicomponent microparticles by integrating all four semiconductors into a single particle. Based on the structural data of the binary and ternary combinations, we presented a modified seeded growth route to synthesize quaternary microparticles, in which **12** ribbons were used as seeds and **34** as the second growth unit (Fig. 4a and Supplementary Fig. 16e, f). Considering that **12** ribbons can afford epitaxial growth of **3** nanorods to form branched heterostructures, we infer that the quaternary microparticles still adopt such a heterostructured architecture using the seed-mediated strategy.

For instance, weak orange-emitting or bright red-emitting aligned nanorods are selectively grown on a green-emitting ribbon to form a branched architecture at $y_2 = 1\%$, $x_4 = 0.001\%$

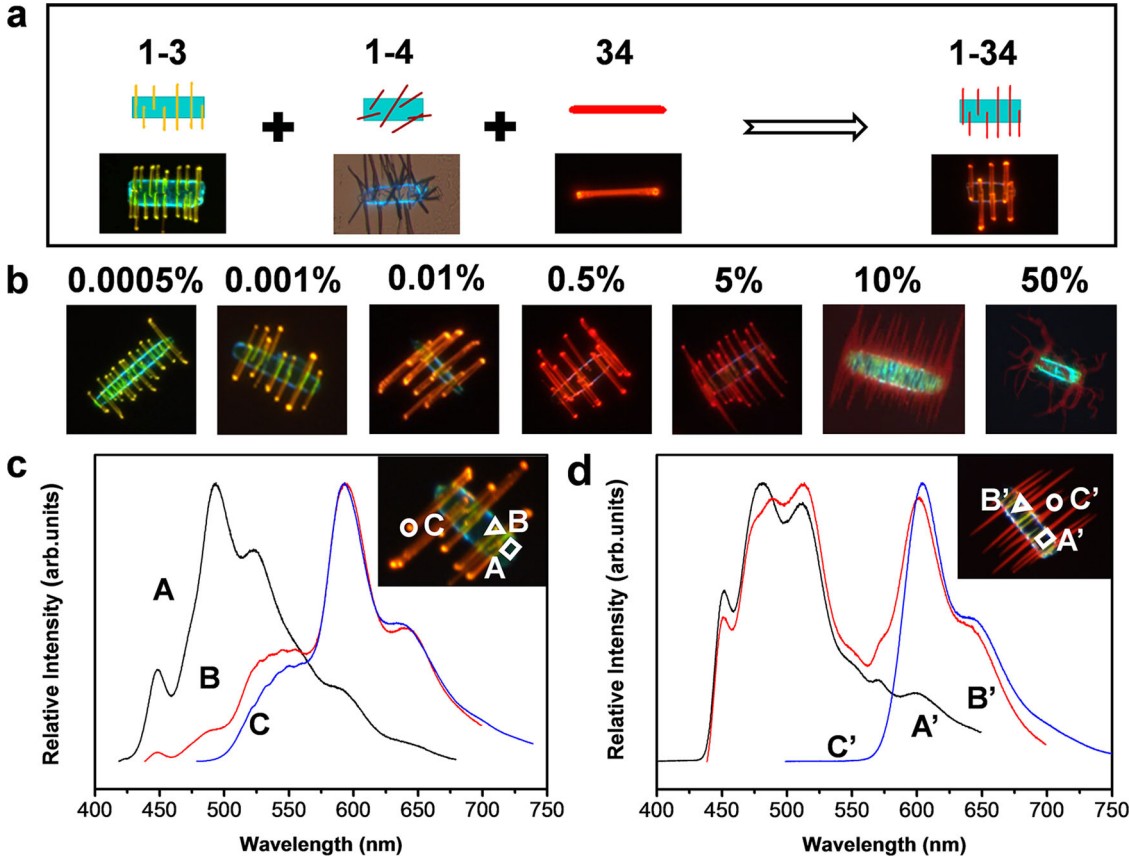

**Fig. 3 Rational construction of ternary microparticles made of 1, 3, and 4 based on the miscibility of two-component combinations. a** Schematic showing the design principle of **1-34** branched heterostructure (top row is a color-coded diagram of the expected particle; bottom row is a fluorescence microscopy image of the resultant particle). **b** Fluorescence microscopy images of **1-34** branched heterostructures with a variable molar ratio of **3/4**. **c, d** Microarea PL spectra of single **1-34** branched heterostructures formed at $x_4 =$ (**c**) 0.01% and **d** 5%. The insets show the corresponding fluorescence microscopy images.

or $y_2 = 5\%$, $x_4 = 5\%$ (Fig. 4b). By means of microarea PL spectra (Fig. 4c), we can clearly recognize the composition and emission color of each domain for single quaternary microparticle ($y_2 = 1\%$, $x_4 = 0.5\%$). Notably, PL of location **D** corresponds to green emission from **12** due to ET from **1** to **2**, whereas that of location **F** corresponds to red emission from **34** due to ET from **3** to **4**. Instead, disordered helical ribbons with red emission rather than straight aligned rods were attached on a green-emitting ribbon to form quaternary heterostructures at $y_2 = 10\%$, $x_4 = 50\%$ (Supplementary Fig. 19), again proving that the architectures of the quaternary combinations are highly composition-dependent.

To demonstrate the efficacy of the present synthetic strategy, we further checked the cases by observing fluorescence microscopy results of the as-prepared multicomponent microparticles over a wide scale range (Supplementary Fig. 20). Similarly to the single type particles (Fig. 1b–e), multitype particles formed with good structural uniformity and consistency without the necessity of purification. Note that a low supersaturation of the second growth component is required for the formation of branched heterostructures, which facilitates its heterogeneous nucleation on the seed ribbons. Importantly, the synthetic methodologies are extremely simple and highly reproducible regardless of whether the multicomponent particles are formed in an alloyed and/or a phase separated state.

The rational construction of multicomponent particle libraries has clearly revealed that the configurations and spatial distribution of multicomponent particles are mainly determined by structural compatibility and molar ratio among multiple components, which are largely independent of their synthetic procedures and growth dynamics. After stored in the dark for half a year, the fluorescence microscopy images of some representative multicomponent microparticles (Supplementary Fig. 21) were again examined, and we found that the microparticles ranging from unary to quaternary components have no obvious degradation or damages, confirming their good long-term stability. Hence, the miscibility of the components may be considered as a general design basis to direct the synthesis of organic multicomponent particles beyond three components, that are hard to predict by conventional methods.

**Binary, ternary, and quaternary microparticles incorporating core–shell configuration.** The successes in precise morphology synthesis and elaborate composition control for the above multicomponent microparticles provides us a powerful tool to construct binary, ternary, and quaternary microparticles with more complex architectures (Fig. 5a and Supplementary Fig. 22). For instance, uniform coverage of **3** seed microrods with **34** alloy to form **3**-**34** core–shell microstructures was realized by a two-step seeded growth route. As shown in Fig. 5b, c, the fluorescence microscopy results clearly reveal the formation of yellow-red dual-color-emitting **3**–**34** core–shell heterostructures, where **3** microrods serve as seeds to afford the growth of **34** alloy layer. We further extended the library of multicomponent complex microparticles through such a seed-mediated strategy by using **1**–**3** or **12**–**3** branched heterostructure as a seed (Fig. 5a). As

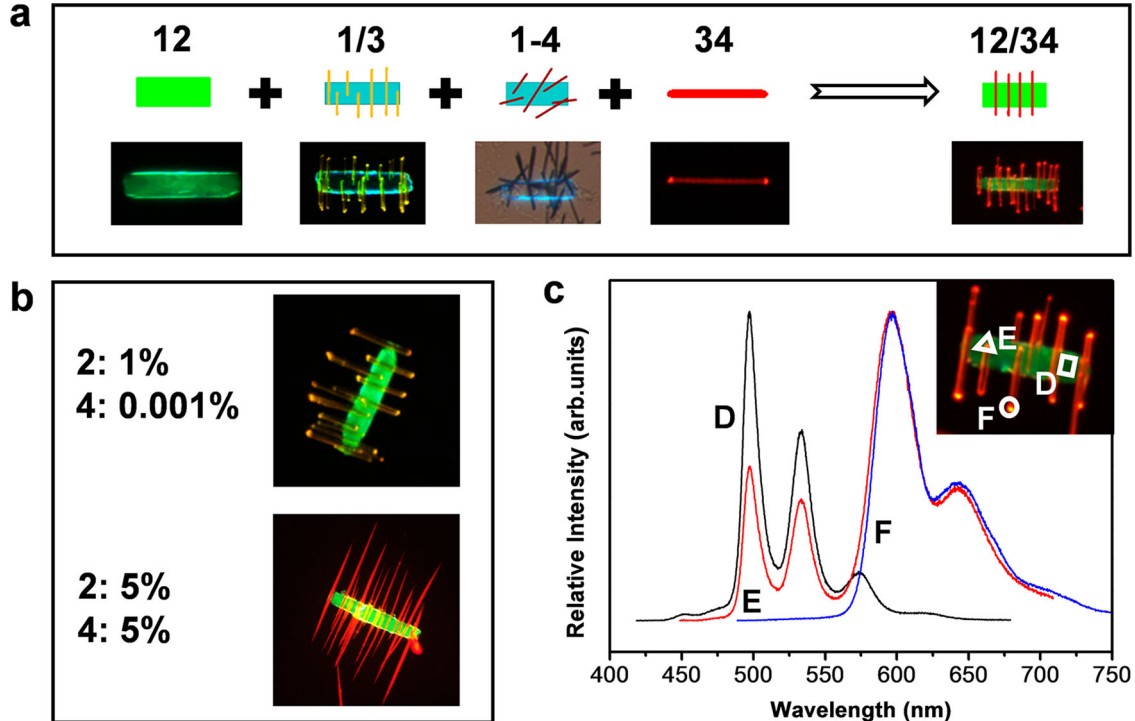

**Fig. 4 Rational construction of quaternary microparticles made of 1, 2, 3, and 4 based on the miscibility of two-component combinations. a** Schematic showing the design principle of **12**-**34** branched heterostructure (top row is a color-coded diagram of the expected particle; bottom row is a fluorescence microscopy image of the resultant particle). **b** Fluorescence microscopy images of **12**-**34** branched heterostructures with a variable molar ratio of **1/2** and **3/4** (top row: $y_2 = 1\%$, $x_4 = 0.001\%$; bottom row: $y_2 = 5\%$, $x_4 = 5\%$). **c** Microarea PL spectra of single **12**-**34** branched heterostructure with a specific molar ratio of **1/2** and **3/4** ($y_2 = 1\%$, $x_4 = 0.5\%$). The inset shows the corresponding fluorescence microscopy image.

expected, the corresponding confocal laser fluorescence microscopy images (CLFM) collected from (Fig. 5e) blue-light, (Fig. 5f) yellow-light, and (Fig. 5g) red-light regions as well as (Fig. 5d) the tricolor superposed CLFM image confirm that the **3** nanorod branches were coated by **34** alloy layer upon employing **1**–**3** branched heterostructure as a seed. On the basis of the miscibility of the four compounds, **12**-**3**-**34** microparticles has also been realized readily (Fig. 5h–k).

**Optical waveguiding properties of binary alloy microparticles.** The successful synthesis of multicomponent microparticles comprising organic luminescent materials makes it possible to integrate these architectures into miniaturized photonic devices. As a proof of concept, we studied the optical waveguiding properties of binary alloy assemblies of **34** ($x_4 = 0$, 0.001, 0.5, and 5%) through a home-made micro-PL setup (Supplementary Fig. 23). Under excitation with a 405 nm laser, bright waveguide spots on both the two tips of these 1D alloy assemblies can be clearly observed (Supplementary Fig. 24a, c, e, g), indicating excellent photon transport behavior of **34**. For single pure **3** microrod ($x_4 = 0$), the out-coupled PL spectra from its one tip have been collected by moving the excitation spot, as presented in Supplementary Fig. 24b. Typically, spatially resolved PL peaks (550 and 590 nm) can be revealed and their waveguide losses were further analyzed based on the functional relationships of PL intensity ratio of the excitation spot ($I_{ex}$) and the emitting tip ($I_{tip}$) and propagation distance ($d$). As a result, the ratio from these two PL peaks shows single-exponentially decay as the increase of the propagation distance (inset in Supplementary Fig. 24b). According to a single-exponential function (1), optical loss coefficient ($\alpha$, dB mm$^{-1}$) of guided light is determined

$$I_{ex}/I_{tip} = Ae \wedge (-\alpha d), \qquad (1)$$

where $A$ is a constant. The $\alpha$ values for pure **3** microrod at 550 and 590 nm were calculated to be 32 ad 29.5 dB mm$^{-1}$, respectively.

Notably, the loss coefficient of **34** alloy microrod at $x_4 = 0.001\%$ increases by 15–17 dB mm$^{-1}$ at around 555 and 596 nm (Supplementary Fig. 24d). As described above, the emission colors of **34** alloys would shift towards red ($x_4 = 0.5\%$, Supplementary Fig. 24e) and dark red light ($x_4 = 5\%$, Supplementary Fig. 24g), respectively, upon increasing the content of **4**. Unexpectedly, the waveguide losses at longer wavelengths ($\lambda_{em} > 600$ nm) would decrease dramatically and the coefficient $\alpha$ is calculated to be below 20 dB mm$^{-1}$ (Supplementary Table 3). At low $x_4$ ($x_4 = 0.001\%$), the PL spectra of **34** alloys exhibit highly similar PL profiles to pure **3** except the variation of intensity ratio of PL peaks, thus leading to dramatic re-absorption effect and then large optical loss. As $x_4$ increases ($x_4 = 0.5$ and 5%), effective separation of the absorption of the energy donor (**3**) from the PL of the energy acceptor (**4**) can be enabled via ET from **3** to **4**. Such a process named remote energy relay (RER)[54] in **34** alloy assemblies at relatively high $x_4$ can greatly reduce the optical loss, thus enabling more superior photon transport abilities relative to pure **3** microrods. The optical waveguide behavior of binary alloy assemblies can be controllably modulated by varying the stoichiometric ratio, confirming their promising potential as photonic devices. The rational and elaborate control of highly integrated multicomponent organic microparticles with up to four compounds represents significant improvements in synthetic methodology, which would be beneficial for the realization of desired photonic and electronic applications.

**Discussion**

Similar to the cases of multimetallic nanoparticles, organic analogs comprising multiple organic semiconductors can be

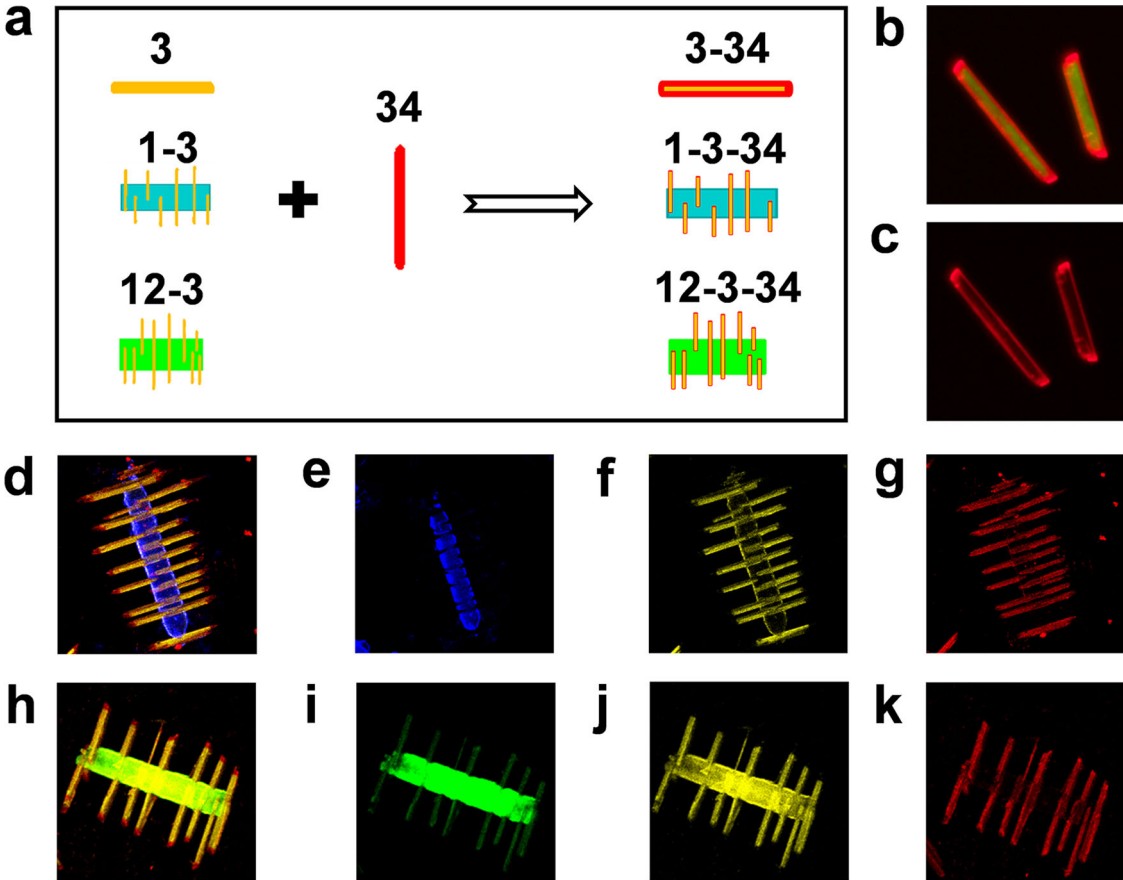

**Fig. 5 Rational construction of binary, ternary, and quaternary microparticles incorporating core–shell configuration. a** Schematic showing the design principle of **3-34**, **1-3-34**, and **12-3-34** heterostructures. **b**, **c** Fluorescence microscopy images of **3-34** ($x_4 = 50\%$) core-shell heterostructures under the excitation of **b** UV and **c** green light. **d–g**, CLFM images of single **1-3-34** ($x_4 = 50\%$) heterostructure collected from **e** blue-light, **f** yellow-light, and **g** red-light regions. **d** The corresponding tricolor superposed CLFM image. **h–k** CLFM images of single **12-3-34** ($y_2 = 10\%$, $x_4 = 50\%$) heterostructure collected from **i** green-light, **j** yellow-light, and **k** red-light regions. **h** The corresponding tricolor superposed CLFM image.

constructed in a rational manner by one-pot or stepwise solution-based route. The architectures of each material combination of these organic molecules are in programming control and can be predicted in advance, based on tailorable structural compatibility among different constituent materials. The efficiency to synthe-size such multicomponent microparticle libraries provides a promising platform to investigate an alloyed state or phase separation. Given the structural diversity of organic molecules as well as different mixing behavior, composition, shape, and orientation of each domain of organic microparticles can be further adjusted and modified, giving rise to multicomponent architectures with growing complexity. We can envisage that more material combinations exceeding thirty species would be achieved upon combining the fifth component, such as pentacene to the quaternary systems. As such, functional π-conjugated organic semiconductors can be specifically designed and synthesized to achieve desired multicomponent architectures, which could exhibit broad performance applications ranging from multicolor lasers, fluorescent barcodes, to light-emitting diodes.

## Methods

**Materials**. *p*-Distyrylbenzene (**1**), tetracene (**2**), 9,10-bis(phenylethynyl)anthracene (**3**), and 5,12-bis(phenylethynyl)naphthacene (**4**) were purchased from Sigma-Aldrich, and used without further treatment. THF (A.R.), cyclohexane (A.R.), CHCl₃ (A.R.), and ethanol (HPLC) were purchased from Beijing Chemical Agent Ltd., China. Ultrapure water with a resistivity of 18.2 MΩ cm⁻¹ was obtained by using a Milli-Q apparatus (Milipore).

**Synthesis of unary microparticles of four organic semiconductors**. Typically, pure assemblies of **1** were prepared by slow evaporation of a drop of a monomer solution of **1** in THF/cyclohexane ($C_1 = 5$ mM, $v/v = 1:3$). Also, pure assemblies of **2** were achieved by fast mixing a monomer solution of **2** in THF with cyclohexane ($C_2 = 5$ mM, $v/v = 1:1$). While assemblies of **3** or **4** were formed by mixing 1 mL of its monomer solution in THF ($C_3 = 5$ mM or $C_4 = 5$ mM) with 5 mL of ethanol/ $H_2O$ mixture ($v/v = 4:1$), respectively.

**Synthesis of binary microparticles of four organic semiconductors**. As a typical example, alloy assemblies of **12** were obtained by slow evaporation of **1/2** solution in THF/cyclohexane ($v/v = 1:2$) with different molar ratio ($C_1 = 5$ mM, $y_2 = 0.1, 1,$ 10%). While alloy assemblies of **34** were synthesized by fast injecting 1 mL of a monomer solution containing **3/4** in THF with different molar ratio ($C_3 = 5$ mM, $x_4 = 0.0005, 0.001, 0.01, 0.5, 5, 10,$ and 50%) into 5 mL of an ethanol/$H_2O$ mixture ($v/v = 4:1$) following previous preparation procedures.

A two-step seeded growth method was used to obtain **1–3** branched heterostructures using **1** ribbons as seeds. Specifically, the as-prepared **1** microribbons were immersed in a monomer solution of **3** in CHCl₃/ethanol ($C_3 = 2$ mM, $v/v = 1:1$) and then the suspension was slowly dried at room temperature. Similarly, **2–3** branched heterostructures were synthesized by immersion of the preformed **2** ribbons in a monomer solution of **3** in CHCl₃/ethanol ($C_3 = 2$ mM, $v/v = 1:1$), and then slow evaporation of the suspension. Binary heterostructures with ill-defined morphologies involving **1–4** and **2–4** were also formed following the above stepwise seeded growth strategy.

**Synthesis of ternary microparticles of four organic semiconductors**. As a typical example, **1–34** ternary branched heterostructures were obtained by immersion of the pre-existing **1** microribbons in precursor solutions of **3/4** in CHCl₃/ethanol ($v/v = 1:1$) with a variable molar ratio ($C_3 = 2$ mM, $x_4 = 0.0005,$ 0.001, 0.01, 0.5, 5, 10, and 50%), and then slow evaporation of the suspensions at room temperature. Moreover, the other two branched heterostructures involving **12-3** ($C_1 = 5$ mM, $y_2 = 5\%$, $C_3 = 2$ mM) and **2–34** ($C_2 = 5$ mM, $C_3 = 2$ mM, $x_4 =$

0.5%) and disordered **12-4** heterostructures ($C_1 = 5$ mM, $y_2 = 1\%$, $C_4 = 2$ mM) were also synthesized following similar preparation procedures.

**Synthesis of quaternary microparticles of four organic semiconductors.** Typically, **12–34** quaternary branched heterostructures were obtained by immersion of the pre-existing **12** microribbons with a variable molar ratio ($C_1 = 5$ mM, $y_2 = 1, 5\%$) in precursor solutions of **3/4** in CHCl₃/ethanol ($C_3 = 2$ mM, $x_4 = 0.001, 0.5,$ and $5\%,$ $v/v = 1{:}1$) and then slow evaporation of the suspensions at room temperature.

**Synthesis of single crystals of 1 and 4.** Single crystal of pure **1** was grown by slow evaporation of a stock solution of **1** in THF/1,4-dioxane/cyclohexane stored in a sealed pipe ($C_1 = 2$ mM, $v/v/v = 12{:}8{:}1$). Similarly, single crystal of **4** was formed by slow evaporation of a stock solution of **4** in THF ($C_4 = 5$ mM).

**Characterization.** The morphologies and sizes of the samples were measured using field-emission scanning electron microscopy (FESEM, Hitachi S4800) at an acceleration voltage of 5 kV. Prior to analysis, the samples were coated with a thin gold layer using an Edwards Sputter Coater. TEM images were obtained using a JEOL-2100F electron microscope operated at 80 kV. One drop of the as-prepared colloidal dispersion was deposited on a carbon-coated copper grid, and dried under high vacuum. The powder X-ray diffraction (PXRD) patterns were measured by a D/max 2400 X-ray diffractometer with Cu Kα radiation ($\lambda = 1.54050$ Å) operated in the 2θ range from 3 to 40°, by using the samples spin-coated on the surface of a quartz substrate. And, the single crystal X-ray diffraction (SCXRD) patterns for compounds **1** and **4** were collected at 293 K on a Rigaku Oxford Diffraction Supernova Dual Source equipped with an AtlasS2 CCD using Cu Kα ($\lambda = 1.54184$ Å for **1**) radiation or Mo K ($\lambda = 0.71073$ Å for **4**) radiation. The data was collected and processed using CrysAlisPro. The structures were solved by direct methods using Olex2 software and the non-hydrogen atoms were located from the trial structure, and then refined anisotropically with SHELXL-2018 using a full-matrix least squares procedure based on $F^2$. The PL spectra of the samples were measured on a HORIBA JOBIN YVON FLUOROMAX-4 spectrofluorimeter with a slit width of 1 nm. The samples were both deposited on the surface of a quartz substrate. The fluorescence microscopy images were obtained using a Leica DMRBE fluorescence microscopy with a spot-enhanced charged couple device (CCD, Diagnostic Instrument, Inc.). The samples were prepared by placing a drop of dispersion onto a cleaned quartz slide. The photoluminescence quantum yields (PLQYs, Φ) were measured on an Edinburgh FLS920 spectrometer with an integrating sphere. Microarea PL spectra of each domain of branched nanorod heterostructures were collected using laser confocal scanning microscopy (Leica, TCS-SP5) equipped with an UV laser (405 nm). Single alloy assemblies of **34** dispersed on a glass substrate were excited locally at different position with a 400 nm laser from the Tisapphire laser with a BBO crystal (~150 fs, 1 KHz) vertically focused down to a ~10 μm diameter spot through an objective (Nikon CFLU Plan). The excitation laser was filtered with a 405 nm bandpass filter.

## Data availability

The data that support the findings of this study are available from the authors on reasonable request. The X-ray crystallographic coordinates for compounds **1** and **4** reported in this study have been deposited at the Cambridge Crystallographic Data Centre (CCDC) under deposition numbers 2031306 and 2031319, while those for compounds **2** and **3** were obtained from CCDC under deposition numbers 114446 and 715257. These data can be obtained free of charge from CCDC via www.ccdc.cam.ac.uk/data_request/cif.

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

## Acknowledgements

This work was supported by the Ministry of Science and Technology of China (Grant No. 2017YFA0204503), the National Natural Science Foundation of China (Grant Nos. 21971189 and 21790364).

## Author contributions

Y.L.L. and H.B.F. directed and supervised the project. Y.L.L. designed all experiments. D.D.Z. carried out the experiments, characterization and collected the data. J.B.D. performed the optical waveguide measurements. Y.L.L. wrote the manuscript and all authors have agreed to the content of the manuscript.

## Competing interests

The authors declare no competing interests.
