## [Peer Review File · Nature Communications]

REVIEWER COMMENTS

Reviewer #1 (Remarks to the Author):

Building further on a an approach, developed by the authors, they now mix up till 4 organic dyes in microparticles, and tune the morphology and composition of these nanoparticles. The novelty of this manuscript relates to the number of organic dyes that are combined to form nanoparticles, and to the control of the morphology and spectroscopic properties, based on the (lack of) miscibility of the dyes involved (ranging from alloys to phase separated crystals). Mixing up till 4 nanoparticles seems indeed to be a clear novelty. The authors could follow the growth of these microparticles, and achieve a good level of control by tuning the mixing protocol. Though this work seems a natural extension to me of work done by these authors before, in my opinion, the main achievement is the number of components involved, and the level of control. As such, I believe it may be at the level of impact expected for a journal such as Nature Communications.

What I miss though is some kind of application (proof of concept, even preliminary data), indicating that indeed this level of control is beneficial for some of the applications mentioned in the conclusions.

Furthermore, I missed somehow statements on the (long term) stability of these particles, and a discussion of the kinetics and thermodynamics involved.

Reviewer #2 (Remarks to the Author):

Zhang et al. described a crystal engineering process of incorporating multiple molecular components into one single structure and compared the significance of such a process with inorganic metal alloys. While I found the work of aesthetic values, I have to disagree with the authors' claims. The work did not demonstrate any new findings other than expected doping (exciton energy transfer) and/or phase-separated behaviors among different molecular domains. The contrast to inorganic alloys is that their electronic state has been completely altered which can only be explained by quantum mechanics; the organic counterparts are mostly non-interacting systems (or weakly interacting ones at best) with no new physics or chemistry added. If the authors are to market the current work as useful application precedents, they need to demonstrate instead of merely speculating that "which could exhibit broad performance applications ranging from multicolor lasers, fluorescent barcodes, to light-emitting diodes". After all, the work is of purely engineering nature without significant scientific perspective. Therefore, my recommendation is that either the author show how they can achieve what they claim in the end by using these molecules or they submit it to somewhere more specialized such as CrystEngComm.

Reviewer #3 (Remarks to the Author):

In this work, Prof. Fu and coworkers prepared several new multiple organic semiconductors in a

rational manner by one-pot or stepwise solution-based route. The architectures of each material combination of these organic molecules are in programming control and could be predicted in advance, based on tailorable structural compatibility among different constituent materials. The efficiency to synthesize such multicomponent microparticle libraries provides a promising platform to investigate an alloy state or phase-separation. Given the structural diversity of organic molecules as well as different mixing behavior, composition, structure, and shape of each domain of organic microparticles can be further adjusted and modified, which might be very useful for machine learning to explore new functional materials. The results shown is very interesting and should be accepted for publication after minor revision. 1) The authors should provide more detail description about basic structures or energy levels match for the selection of effective component units. The authors should discuss the efficiency of the synthesis, if they indicated the process as 'library'. Of course, the preparation at now stage was still be working in the solutions, which might need purification. If so, this approach lefts one step to achieve real 'one-pot' preparation for functional materials. 2) In fact, co-assembling approach indeed produce some new luminescent materials. Please see and cite some examples: J. Fraser Stoddart, et al. *J. Am. Chem. Soc.*, 2020, 142, 16849-16860; D. Qu, et al. *Nature Comm.* 2020, 11, 158; X. Ma, et al. *Acc. Chem. Res.* 2019, 52(3), 738-748. Several figures were blurry and should be improved more clearly.

Referee #1:

Building further on an approach, developed by the authors, they now mix up till 4 organic dyes in microparticles, and tune the morphology and composition of these nanoparticles. The novelty of this manuscript relates to the number of organic dyes that are combined to form nanoparticles, and to the control of the morphology and spectroscopic properties, based on the (lack of) miscibility of the dyes involved (ranging from alloys to phase separated crystals). Mixing up till 4 nanoparticles seems indeed to be a clear novelty. The authors could follow the growth of these microparticles, and achieve a good level of control by tuning the mixing protocol. Though this work seems a natural extension to me of work done by these authors before, in my opinion, the main achievement is the number of components involved, and the level of control. As such, I believe it may be at the level of impact expected for a journal such as Nature Communications.

Comments:

1. What I miss though is some kind of application (proof of concept, even preliminary data), indicating that indeed this level of control is beneficial for some of the applications mentioned in the conclusions.

Response:

We really appreciate for the Referee's positive remarks and kind advice on our manuscript. As stated by the Referee 1, rational and elaborate control of highly integrated multicomponent organic microparticles with up to four compounds represents significant improvements in synthetic methodology, which would be beneficial for the realization of novel photonic and electronic applications. To demonstrate the great potential of these complex topological microstructures in optoelectronic applications, we have examined the optical waveguiding properties of binary assemblies of **34** at $x_4 = 0, 0.001\%, 0.5\%,$ and 5% .

Optical waveguiding properties of binary alloy microparticles. The successful synthesis of multicomponent microparticles comprising organic luminescent materials makes it possible to integrate these architectures into miniaturized photonic devices. As a proof of concept, we studied the optical waveguiding properties of binary alloy assemblies of **34** ($x_4 = 0, 0.001\%, 0.5\%, 5\%$) through a home-made micro-PL setup (Supplementary Fig. 23). Under excitation with a 405 nm laser, bright waveguide spots on both the two tips of these 1D alloy assemblies can be clearly observed (Supplementary Fig. 24a, c, e, g), indicating superior photon transport behavior of **34**. For single pure **3** microrod ($x_4 = 0$), the out-coupled PL spectra from its one tip have been collected by moving the excitation spot, as presented in Supplementary Fig. 24b. Typically, spatially resolved PL peaks (550 and 590 nm) can be revealed and their waveguide losses were further analyzed based on the functional relationships of PL intensity ratio of the excitation spot (I_{ex}) and the emitting tip (I_{tip}) and propagation

distance (d). As a result, the ratio from these two PL peaks both show single-exponentially decay as the increase of the propagation distance (inset in Supplementary Fig. 24b). According to a single-exponential function $I_{ex}/I_{tip} = A\exp(-\alpha d)$, the optical loss coefficient (α , dB mm⁻¹) at 550 and 590 nm was calculated to be 32 and 29.5 dB mm⁻¹, respectively.

Notably, the loss coefficient of **34** alloy microrod at $x_4 = 0.001\%$ increases by 15 to 17 dB mm⁻¹ at around 555 and 596 nm (Supplementary Fig. 24d). As described above, the emission colors of **34** alloys would shift towards red ($x_4 = 0.5\%$, Supplementary Fig. 24e) and dark red light ($x_4 = 5\%$, Supplementary Fig. 24g), respectively, upon increasing the content of **4**. Unexpectedly, the waveguide losses at longer wavelengths ($\lambda_{em} > 600$ nm) would decrease dramatically and the coefficient α is calculated to be below 20 dB mm⁻¹ (Supplementary Table 3). At low x_4 ($x_4 = 0.001\%$), the PL spectra of **34** alloys exhibit highly similar PL profiles to pure **3** except the variation of intensity ratio of PL peaks, thus leading to dramatic re-absorption effect and then large optical loss. As x_4 increases ($x_4 = 0.5\%$ and 5%), effective separation of the absorption of the energy donor (**3**) from the PL of the energy acceptor (**4**) can be enabled via ET from **3** to **4**. As a consequence, such a process named remote energy relay (RER)⁵⁴ can greatly improve the photon transport abilities of **34** alloy assemblies, confirming the promising potential of binary alloys as photonic devices. The rational and elaborate control of highly integrated multicomponent organic microparticles with up to four compounds represents significant improvements in synthetic methodology, which would be beneficial for the realization of novel photonic and electronic applications.

54. Liao, Q., Fu, H. B. & Yao, J. N. Waveguide modulator by energy remote relay from binary organic crystalline microtubes. *Adv. Mater.* **21**, 4153–4157 (2009).

Supplementary Figure 23 Schematic demonstration of the experimental setup for the optical waveguiding characterization: (a) the confocal optical microscopy and (b) the transmittance optical path for the waveguide measurements.

Supplementary Figure 24 Bright-field and PL images of 34 alloy microrods formed at $x_4 =$ (a) 0, (c) 0.001%, (e) 0.5%, and (g) 5% by exciting each rod at different positions. Scale bars, 20 μm . (b, d, f, h) The corresponding spatially resolved PL spectra, collected from one tip of each microrod with variable propagation distance d . The insets display the logarithmic relation of I_{tip}/I_{ex} vs d at different PL peaks, where I_{tip} and I_{ex} are the PL intensity at the output tip and the excited spot, respectively.

Supplementary Table 3 Summarized optical loss coefficient of binary **34** alloy assemblies at $x_4 = 0, 0.001\%, 0.5\%,$ and 5% .

	34			
	$x_4 = 0$	$x_4 = 0.001\%$	$x_4 = 0.5\%$	$x_4 = 5\%$
λ_{em} (nm)	550, 590	555, 596	610, 645	650
α (dB mm^{-1})	32.0, 29.5	47.1, 46.1	21.3, 21.2	15.6

2. Furthermore, I missed somehow statements on the (long term) stability of these particles, and a discussion of the kinetics and thermodynamics involved.

Response:

Thanks for the Referee's kind suggestions. To examine their stability, we again performed fluorescence microscopy tests of representative multicomponent microparticles involving **3**, **1-3**, **34** ($x_4 = 50\%$), **1-34** ($x_4 = 50\%$), **2-34** ($x_4 = 5\%$), **12-34** ($x_4 = 0.001\%$, $y_2 = 10\%$), **12-34** ($x_4 = 5\%$, $y_2 = 0.1\%$), **12-34** ($x_4 = 5\%$, $y_2 = 5\%$), **12-34** ($x_4 = 20\%$, $y_2 = 10\%$), **3-34** ($x_4 = 50\%$), **1-3-34** ($x_4 = 50\%$), and **12-3-34** ($x_4 =$

50%, $y_2 = 10%$) after storage for half a year, as shown in Supplementary Fig. 21.

As depicted by the Referee, we paid little attention to growth dynamics of multicomponent particles in the present work. However, structural and spectrum analysis has revealed that the configurations and composition distribution of multicomponent particles over unary component are mainly determined by compatible molecular structures (molecular shapes, sizes, packing motifs, and intermolecular interactions) and/or closely related epitaxial crystallographic parameters among multiple components, which are largely independent of their growth dynamics. Instead, special concern may be paid to the growth dynamics of the seed crystals, especially for polymorphic seed compounds, considering the polymorphism of π -conjugated molecules.

Supplementary Figure 21 Fluorescence microscopy images of some representative multicomponent microparticles involving (a) **3**, (b) **1-3**, (c) **34** ($x_4 = 50%$), (d) **1-34** ($x_4 = 50%$), (e) **2-34** ($x_4 = 10%$), (f) **12-34** ($x_4 = 0.001%$, $y_2 = 10%$), (g) **12-34** ($x_4 = 5%$, $y_2 = 0.1%$), (h) **12-34** ($x_4 = 5%$, $y_2 = 5%$), (i) **12-34** ($x_4 = 20%$, $y_2 = 10%$), (j) **3-34** ($x_4 = 50%$), (k) **1-3-34** ($x_4 = 50%$), and (l) **12-3-34** ($x_4 = 50%$, $y_2 = 10%$) after stored in the dark for half a year. Scale bars, 10 μm .

After stored in the dark for half a year, the fluorescence microscopy images of some representative multicomponent microparticles (Supplementary Fig. 21) were again examined and we found that the microparticles ranging from unary to quaternary

components have no obvious degradation or damages, confirming their good long-term stability.

The rational construction of multicomponent particle libraries has clearly revealed that the configurations and spatial distribution of multicomponent particles are mainly determined by structural compatibility and composition range among multiple components, which are largely independent of their synthetic procedures and growth dynamics.

Referee #2:

Zhang et al. described a crystal engineering process of incorporating multiple molecular components into one single structure and compared the significance of such a process with inorganic metal alloys. While I found the work of aesthetic values, I have to disagree with the authors' claims. The work did not demonstrate any new findings other than expected doping (exciton energy transfer) and/or phase-separated behaviors among different molecular domains. The contrast to inorganic alloys is that their electronic state has been completely altered which can only be explained by quantum mechanics; the organic counterparts are mostly non-interacting systems (or weakly interacting ones at best) with no new physics or chemistry added. If the authors are to market the current work as useful application precedents, they need to demonstrate instead of merely speculating that "which could exhibit broad performance applications ranging from multicolor lasers, fluorescent barcodes, to light-emitting diodes". After all, the work is of purely engineering nature without significant scientific perspective. Therefore, my recommendation is that either the author show how they can achieve what they claim in the end by using these molecules or they submit it to somewhere more specialized such as CrystEngComm.

Response:

We appreciate the Referee's upright remarks and useful advice on our manuscript. As described by the Referee 2, we focus on synthetic methodologies of multicomponent organic microparticles in previous manuscript. To further reveal the application potential of these multicomponent organic microparticles, we have added the optical waveguiding measurements of binary alloy assemblies of **34** at variable molar ratio (See Response 1 to Referee 1). We believe that the rational and elaborate control of highly integrated multicomponent organic microparticles with up to four compounds not only represents significant improvements in synthetic methodology, but also would be beneficial for the realization of novel photonic and electronic applications.

Referee #3:

In this work, Prof. Fu and coworkers prepared several new multiple organic semiconductors in a rational manner by one-pot or stepwise solution-based route. The

architectures of each material combination of these organic molecules are in programming control and could be predicted in advance, based on tailorable structural compatibility among different constituent materials. The efficiency to synthesize such multicomponent microparticle libraries provides a promising platform to investigate an alloy state or phase-separation. Given the structural diversity of organic molecules as well as different mixing behavior, composition, structure, and shape of each domain of organic microparticles can be further adjusted and modified, which might be very useful for machine learning to explore new functional materials. The results shown is very interesting and should be accepted for publication after minor revision.

Comments:

1) The authors should provide more detail description about basic structures or energy levels match for the selection of effective component units. The authors should discuss the efficiency of the synthesis, if they indicated the process as ‘library’. Of course, the preparation at now stage was still be working in the solutions, which might need purification. If so, this approach leaves one step to achieve real ‘one-pot’ preparation for functional materials.

Response:

Thanks a lot for the Referee’s positive comments and kind advice on our manuscript. For comparison, the molecular packing motifs and intermolecular interactions of **1**, **2**, **3**, and **4** single crystals (Supplementary Fig. 4) were also revealed based on their crystallographic data. Interestingly, all four crystals adopt a typical herringbone structure with herringbone angles of 59.9°, 51.4°, 79.2° and 84.8°, respectively. The molecules in the crystals **1-3** have good planarity, while the molecules in the crystal **4** deviate from planarity with two similar torsion angles (29.5° and 32.5°) between the phenylethynyl group and the naphthacene core. Moreover, there are strong multiple C–H··· π interactions in all four crystals as well as additional π – π stacking interactions in the crystals **3** and **4**.

Supplementary Figure 4 Molecular packing motifs and herringbone angles of four organic single crystals involving **1**, **2**, **3**, and **4** based on their crystallographic data.

Growth mechanism of binary microparticles with different architectures. To understand why such different configurations form in these binary systems, we took **34** ($x_4 = 5\%$) and **1-3** as typical examples and studied the detailed structural relationship between **3** and **4** or between **1** and **3**. As described above, it has been clearly revealed that compounds **3** and **4** were both arranged in a slipped π - π packing pattern (Supplementary Fig. 4c and d), due to their highly similar π -conjugated molecular skeletons. Moreover, **3** and **4** molecules have nearly identical lengths ($L_3 = 18.332 \text{ \AA}$; $L_4 = 18.361 \text{ \AA}$) and relatively small differences in width ($W_3 = 8.905 \text{ \AA}$; $W_4 = 11.411 \text{ \AA}$), thus facilitating the substitution of **3** by **4** into **34** alloys^{30,31}. By means of the crystallographic data of **3**, we simulated the molecular packing mode of **34** alloys and found that **4** molecules would randomly occupy the lattice sites of **3** host. However, steric hindrance of the twisted phenylethynyl groups of **4** would force the parallel stacking of between anthracene core of **3** and naphthacene core of **4** to distort at a tilted angle, which enables the maximization of their π - π interactions. At higher molar ratio of **3/4** ($x_4 \geq 33.3\%$), large torsion stress drive the shape transition of **34** alloy assemblies from straight rods to helical ribbons. Moreover, the PL spectrum of **3** shows a large overlap with the absorption spectrum of **4**, as presented in Supplementary Fig. 15a. Well-matched energy levels as well as suitable intermolecular distances between **3** and **4** enable high-efficiency energy transfer (ET) process in **34** alloy assemblies, leading to tunable emissions toward **4**³⁰. Notably, compounds **1** and **2** assemblies also possess similar molecular packing patterns and effective energy level matching (Supplementary Fig. 15b), giving rise to color-tunable **12** alloy assemblies^{52,53}.

Supplementary Figure 15 Absorption spectra of the monomer solutions of (a, orange dashed curve) **3**, (a, brown dashed curve) **4**, (b, blue dashed curve) **1**, and (b, green dashed curve) **2** in THF as well as PL spectra of their crystalline assemblies (a, orange solid curve for **3**; a, brown solid curve for **4**; b, blue solid curve for **1**; b, blue solid curve for **2**). The shadow regions in (a) and (b) represent the overlap area of PL spectrum of **3** and absorption spectrum of **4** or that of PL spectrum of **1** and absorption spectrum of **2**.

For **1-3** branched heterostructured systems (Fig. 2f), **1** and **3** with planar molecular configurations have comparable lengths ($L_1 = 17.949 \text{ \AA}$; $L_3 = 18.332 \text{ \AA}$) and distinctly different widths ($W_1 = 4.006 \text{ \AA}$; $W_3 = 8.905 \text{ \AA}$). As a result, **1** and **3** molecules in their respective assemblies are stacked along [010] direction through C–H $\cdots\pi$ and π – π interactions (Supplementary Fig. 4a, 4c, and Fig. 2f), respectively, providing a great possibility to achieve such phase-separated structures rather than homogeneous alloys.

To demonstrate the efficacy of the present synthetic strategy, we further checked the cases by observing fluorescence microscopy results of the as-prepared multicomponent microparticles over a wide scale range (Supplementary Fig. 20). Similar to the cases in single typical particles (Fig. 1), multiple microparticles still remain good structural uniformity and consistency, which behave as organic multicolor microparticle libraries. Noted that low concentration of monomer solution of the second growth component for the formation of branched heterostructures should be requisite to avoid orthogonal self-assembly. Importantly, the synthetic methodologies are extremely simple and highly reproducible regardless of whether the multicomponent particles are formed in an alloy and/or a phase-separation state.

Supplementary Figure 20 (a,b,c,d,f,g,i,k,l,m,n,o) Fluorescence and (e,h,j) bright-field microscopy images of multiple microparticles involving (a) 1, (b) 2, (c) 3, (d) 4, (e) 4, (f) 12 ($y_2 = 5\%$), (g) 1-3, (h) 1-4, (i) 2-3, (j) 2-4, (k) 34 ($x_4 = 0.01\%$), (l) 12-3 ($y_2 = 5\%$), (m) 1-34 ($x_4 = 10\%$), (n) 2-34 ($x_4 = 0.5\%$), and (o) 12-34 ($x_4 = 0.5\%$, $y_2 = 10\%$) over a wide scale range. Scale bars, 10 μm . Among them, the particles 1, 2, 3, 12, 1-3, 2-3, 34, 12-3, 1-34, 2-34 and 12-34 were excited by UV light; while the particle 4 (d) was excited by green light.

2) In fact, co-assembling approach indeed produce some new luminescent materials. Please see and cite some examples: J. Fraser Stoddart, et al. *J. Am. Chem. Soc.*, 2020, 142, 16849-16860; D. Qu, et al. *Nature Comm.* 2020, 11, 158; X. Ma, et al. *Acc. Chem. Res.* 2019, 52(3), 738-748. Several figures were blurry and should be improved more clearly.

Response:

Thanks for the Referee's kind suggestion. We have added the related references into our manuscript according to the Referee's advice. In addition, we have also greatly improved the quality of several figures in the manuscript and supporting information, such as Figs. 1c, 1d, 2d, and Supplementary Figs. 9, 14.

22. Wu, H., Wang, Y., Jones, L. O., Liu, W., Song, B., Cui, Y., Cai, K., Zhang, L., Shen, D., Chen, X.-Y., Jiao, Y., Stern, C. L., Li, X., Schatz, G. C. & Stoddart, J. F. Ring-in-ring(s) complexes exhibiting tunable multicolor photoluminescence. *J. Am. Chem. Soc.* **142**, 16849–16860 (2020).

23. Wang, Q., Zhang, Q., Zhang, Q.-W., Li, X., Zhao, C.-X., Xu, T.-Y., Qu, D.-H. & Tian, H. Color-tunable single-fluorophore supramolecular system with assembly-encoded emission. *Nat. Commun.* **11**, 158 (2020).

24. Ma, X., Liu, J. & Tian, H. Assembling-induced emission: an efficient approach for amorphous metal-free organic emitting materials with room-temperature phosphorescence. *Acc. Chem. Res.* **52**, 738–748 (2019).

Fig. 1: Microparticle libraries of four organic semiconductors from unary to quaternary microparticles formed via solution-based assembly routes. **a**, Schematic showing the molecular structures of four organic semiconductors involving **1** (blue), **2** (green), **3** (orange), and **4** (red). **b**, Unary microparticles (top row is a color-coded diagram of the expected particle; bottom row is a fluorescence microscopy image of the resultant particle). The particles **1**, **2**, and **3** were excited by UV light, whereas the particle **4** was excited by green light. **c**, Binary microparticles made of every two-component combination of the four semiconductors. Among them, both **12** and **34** represent binary alloys; **1-3**, **1-4**, **2-3**, and **2-4** represent binary heterostructures. **1-4** was excited by UV light combined with bright-field mode, whereas the remaining microparticles were excited by only UV light. **d**, Ternary microparticles made of every three-component combination of the four semiconductors. **e**, A quaternary microparticle made of a combination of **1**, **2**, **3**, and **4**.

Fig. 2: The mixing behavior of binary material combinations involving 3/4 and 1/3. a,d, The detailed growth processes of single binary microparticles involving (a) 34 ($x_4 = 5\%$) ribbon and (d) 1-3 branched heterostructure at different intervals (a, $t = 5, 7, 11,$ and 20 s; d, $t = 30, 60, 65,$ and 68 s). Scale bars, $10\ \mu\text{m}$. b,e, Schematic showing the mixing behavior of (b) 34 ribbons and (e) 1-3 branched heterostructures. c,f, The molecular packing motifs of single (c) 34 ribbon along [010] direction and (f) each domain in 1-3 branched heterostructure along [010] direction.

Supplementary Figure 10 SEM images of binary microparticles involving (a) 12 ($y_2 = 1\%$), (b) 1-3, (c) 1-4, (d) 34 ($x_4 = 0.001\%$), (e) 2-3, (f) 2-4. Scale bars, $10\ \mu\text{m}$.

Supplementary Figure 15 SEM images of ternary microparticles involving (a) **12-3** ($y_2 = 5\%$), (b) **1-34** ($x_4 = 5\%$), (c) **2-34** ($x_4 = 5\%$), (d) **12-4** ($y_2 = 5\%$) and (e,f) quaternary microparticles comprising **1**, **2**, **3**, and **4** at (e) low- and (f) high-magnification. Scale bars, 10 μm .

REVIEWERS' COMMENTS

Reviewer #1 (Remarks to the Author):

The authors have addressed my concern related to providing an "application" (optical waveguiding) of the benefit of the multicomponent particles. Therefore, I have no objections against publication of this manuscript.

Reviewer #2 (Remarks to the Author):

The authors have addressed related problems in the revised manuscript, particularly regarding the application potential of the crystal system. Although I think the current manuscript is in much better shape, the authors ought to justify the advantages of using the mixed microcrystals vs. pure crystals or at least commenting on the scenarios in which one of the two systems is better suited, before the publication of the manuscript.

Reviewer #3 (Remarks to the Author):

This revision should be accepted for publication.

Response Letter

In this document, we provide a point-by-point response to the comments of three reviewers.

Reviewer #1 (Remarks to the Author):

The authors have addressed my concern related to providing an "application" (optical waveguiding) of the benefit of the multicomponent particles. Therefore, I have no objections against publication of this manuscript.

Response:

We really appreciate for the reviewer's positive remarks on our manuscript.

Reviewer #2 (Remarks to the Author):

The authors have addressed related problems in the revised manuscript, particularly regarding the application potential of the crystal system. Although I think the current manuscript is in much better shape, the authors ought to justify the advantages of using the mixed microcrystals vs. pure crystals or at least commenting on the scenarios in which one of the two systems is better suited, before the publication of the manuscript.

Response:

We thank the reviewer for the positive evaluation on our work. We have added brief description on the advantages of using the mixed microcrystals vs. pure crystals following the reviewer's kind advice.

Such a process named remote energy relay (RER) **in 34 alloy assemblies at relatively high x_4 can greatly reduce the optical loss, thus enabling more superior photon transport abilities relative to pure 3 microrods. The optical waveguide behavior of binary alloy assemblies can be controllably modulated by varying the stoichiometric ratio,** confirming their promising potential as photonic devices.

Reviewer #3 (Remarks to the Author):

This revision should be accepted for publication.

Response:

We thank the reviewer for the positive opinion on our work.